# Absence of topological order in the $U(1)$ checkerboard toric code

Maximilian Vieweg⋆, Viktor Kott, Lea Lenke,
Andreas Schellenberger, Kai Phillip Schmidt†

Department of Physics, Staudtstraße 7, Friedrich-Alexander-Universität Erlangen-Nürnberg,
Germany
⋆max.vieweg@fau.de, †kai.phillip.schmidt@fau.de

September 2, 2025

## Abstract

We investigate the $U(1)$ checkerboard toric code which corresponds to the $U(1)$-symmetry enriched toric code with two distinct star sublattices. One can therefore tune from the limit of isolated stars to the uniform system. The uniform system has been conjectured to possess non-Abelian topological order based on quantum Monte Carlo simulations suggesting a non-trivial ground-state degeneracy depending on the compactification of the finite clusters. Here we show that these non-trivial properties can be naturally explained in the perturbative limit of isolated stars. Indeed, the compactification dependence of the ground-state degeneracy can be traced back to geometric constraints stemming from the plaquette operators. Further, the ground-state degeneracy is fully lifted in fourth-order degenerate perturbation theory giving rise to a non-topological phase with confined fracton excitations. These fractons are confined for small perturbations so that they cannot exist as single low-energy excitation in the thermodynamic limit but only as topologically trivially composite particles. However, the confinement scale is shown to be surprisingly large so that gaps are extremely small on finite clusters up to the uniform limit which is calculated explicitly by high-order series expansions. Our findings suggest that these gaps were not distinguished from finite-size effects by the recent quantum Monte Carlo simulation in the uniform limit. All our results therefore point towards the absence of topological order in the $U(1)$ checkerboard toric code along the whole parameter axis.

# 1  Introduction

One of the most important problems in modern physics is building a quantum computer able to perform calculations useful for science and technology. To this end, topologically ordered quantum systems with a ground-state degeneracy stable to local decoherence and elementary excitations with non-Abelian anyonic statistics distinct from bosons and fermions are a fascinating concept. However, building systems hosting topological order remains an enormous challenge.

One theoretical proposal to create such topologically ordered systems is the use of a symmetry principle called combinatorial gauge theory [1–3]. Motivated by this idea, the $U(1)$-symmetry enriched toric code (U1TC) was introduced recently [4]. The U1TC can be viewed as a generalization of the conventional exactly solvable toric code [5] with an additional global $U(1)$ symmetry enforced on the star operators. The conventional toric code is the most paradigmatic microscopic model displaying topological order and Abelian anyons. At the same time it is relevant for quantum error correction in quantum computing devices. In contrast to the conventional toric code, its $U(1)$-symmetry enriched version can not be solved analytically because different star operators do not commute anymore. However, based on finite-temperature and finite-size quantum Monte Carlo simulations (QMC), it has been claimed to display non-Abelian topological order [4], which is the long-sought basis for topological quantum computation. Specifically, numerical results point towards a non-trivial dependence of the ground-state degeneracy on the compactification, which is a clear sign of UV/IR mixing, and a three-fold topological ground-state degeneracy for one of the studied compactifications. Further, it has been shown that the system exhibits weak Hilbert space fragmentation [4,6,7]. Systems with this property are known to violate the strong version of the eigenstate thermalization hypothesis, which was initially observed in systems with fractons [8,9]. Fractons are excitations with restricted mobility that were first recognized in systems generalizing intrinsic topological quantum order to three spatial dimensions [10–15].

In this work, we introduce the checkerboard U1TC with two types of distinct star sublattices following our recent work on the XY toric code (XYTC) [16]. We demonstrate that the checkerboard U1TC is exactly solvable in the limits of isolated star sublattices. Interestingly, one can naturally explain the dependence of the ground-state degeneracy on the compactification by the presence of geometrical constraints enforced by the plaquette operators. Further, fourth-order degenerate perturbation theory results in a non-degenerate unique ground state that does not display topological order. Perturbation theory and exact diagonalization do not give any evidence for a quantum phase transition up to the uniform U1TC. At the same time energy gaps are shown to be extremely small by high-order series expansions. Interestingly, these small energy scales can be traced back to a large confinement scale of fracton excitations which cannot exist as single low-energy excitation in the thermodynamic limit but only as topologically trivially composite particles. Such gaps have not been distinguished from finite-size effects by the recent QMC simulation at finite temperature in the uniform limit [4].

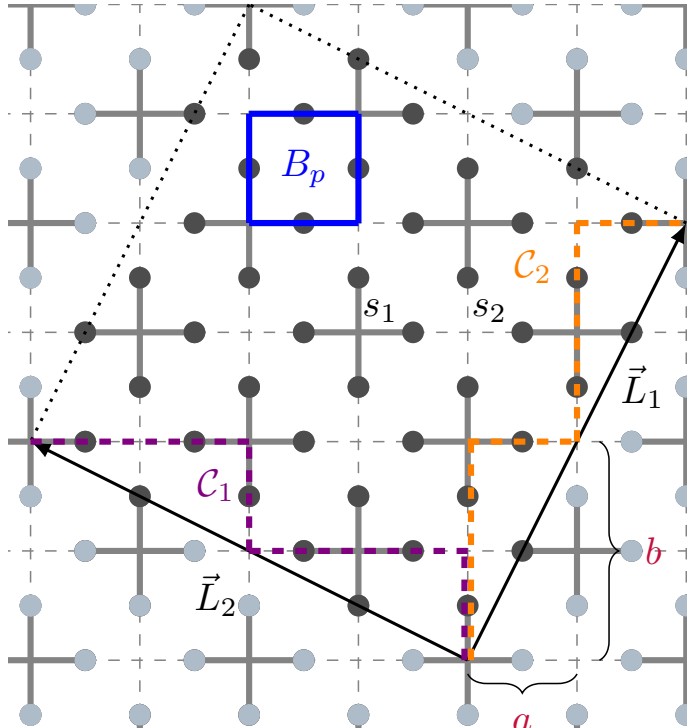

Figure 1: Illustration of the lattice and operators of the checkerboard U1TC. Specifically, a lattice with $L = 2$ corresponding to a compactification characterized by $a = 1$ and $b = 2$ is shown, as defined around Eq. (4) in the main text. Stars in the sublattices $s_1$ and $s_2$ are depicted by solid and dashed stars, respectively. Two non-contractible loops $\mathcal{C}_1, \mathcal{C}_2$ are shown by dashed purple and orange lines.

Consequently, our findings point towards the absence of topological order in the checkerboard U1TC along the full parameter line including the uniform case.

Our article is organized as follows. In Sec. 2 we introduce the checkerboard U1TC and describe its elementary properties. In Sec. 3 we discuss the exactly solvable limit of the checkerboard U1TC. This includes the ground-state degeneracy and its dependence on the compactification in Subsec. 3.1 as well as the star and plaquette excitations in Subsec. 3.2. For the latter we discuss the different types of excitations of the checkerboard U1TC corresponding to confined fracton excitations. In the subsequent Sec. 4 we approach the uniform U1TC by high-order series expansions and exact diagonalization. A conclusion is given in Sec. 5.

## 2   Checkerboard U1TC

The Hamiltonian of the checkerboard U1TC [4] is given by

$$H_{\text{CBU1TC}} = -\sum_{s_1} \tilde{A}_{s_1} - \lambda \sum_{s_2} \tilde{A}_{s_2} - \sum_p B_p , \tag{1}$$

with $B_p = \prod_{i \in p} \sigma_i^z$ and

$$\tilde{A}_{s_1/s_2} = \sigma_1^+ \sigma_2^+ \sigma_3^- \sigma_4^- + \sigma_1^+ \sigma_2^- \sigma_3^+ \sigma_4^- + \sigma_1^+ \sigma_2^- \sigma_3^- \sigma_4^+ + \text{h.c.} . \tag{2}$$

Here we use $\sigma_i^\pm = \frac{1}{2}(\sigma_i^x \pm i\sigma_i^y)$ and $\sigma_i^\alpha$ with $\alpha \in \{x, y, z\}$ the Pauli matrices acting on site $i$. The checkerboard U1TC is depicted in Fig. 1. The parameter $\lambda$ allows to tune from the limit

of isolated stars $s_1$ for $\lambda = 0$ up to the uniform U1TC for $\lambda = 1$, which has been investigated recently by QMC simulations [4]. The checkerboard U1TC explicitly breaks the translational symmetry for $\lambda < 1$, while the symmetry is spontaneously broken for $\lambda = 1$ according to QMC.

As outlined in Ref. [4], the checkerboard U1TC has a global $U(1)$ symmetry associated with the total magnetization $M_z = \sum_i \sigma_i^z$ as a conserved quantity and a local $\mathbb{Z}_2$ symmetry per plaquette $p$ since the $B_p$ operators with eigenvalues $b_p = \pm 1$ commute with the Hamiltonian. Analogous to the conventional toric code, the Wilson loop operators $W_{1/2} = \prod_{i \in \mathcal{C}_{1/2}} \sigma_i^z$, defined along non-contractible loops $\mathcal{C}_{1/2}$, yield independent $\mathbb{Z}_2$ conserved quantities for periodic boundary conditions. We denote the corresponding four symmetry sectors by the eigenvalues $(w_1, w_2) \in \{(+,+), (+,-), (-,+), (-,-)\}$ of the loop operators. However, unlike the conventional toric code, the U1TC is no longer exactly solvable, since $[\tilde{A}_{s_1}, \tilde{A}_{s_2}] \neq 0$ for neighboring stars $s_1$ and $s_2$. The $\tilde{A}_s$ term is similar to the one of the checkerboard XYTC

$$\tilde{A}_s + \sigma_1^+ \sigma_2^+ \sigma_3^+ \sigma_4^+ + \sigma_1^- \sigma_2^- \sigma_3^- \sigma_4^- = \sqrt{2} A_s^{\text{XYTC}}, \tag{3}$$

where $A_s^{\text{XYTC}}$ is the star operator of the checkerboard XYTC for $\phi = \frac{\pi}{4}$ [16]. The latter possesses exact subsystem symmetries that are spontaneously broken giving rise to symmetry-protected fracton excitations [16]. In the checkerboard U1TC these symmetries are explicitly broken. Finally, let us mention that both star operators $\tilde{A}_s$ and $A_s^{\text{XYTC}}$ can be seen as reduced versions of the star operator of the conventional toric code, by omitting terms with odd numbers of $\sigma_i^+$ or $\sigma_i^-$.

QMC simulations indicate that the uniform U1TC ($\lambda = 1$) displays a non-Abelian topologically ordered ground state [4]. In particular, a non-trivial dependence of the ground-state degeneracy on the compactification is detected. Let us therefore introduce the following two orthogonal compactification vectors

$$\begin{aligned} \vec{L}_1 &= L(a\vec{x} + b\vec{y}), \\ \vec{L}_2 &= L(-b\vec{x} + a\vec{y}), \end{aligned} \tag{4}$$

which are parameterized by two non-negative coprime integers $a$ and $b$ and the horizontal and vertical distance $\vec{x}, \vec{y}$ between neighboring stars. The compactification is then defined by identifying all vectors $\vec{r}$ with $\vec{r} + \vec{L}_1$ and $\vec{r} + \vec{L}_2$ as seen in Fig. 1. The relevant compactifications for this work are $a = 0, b = 1$ (called 0° compactification) and $a = 1, b = 1$ (called 45° compactification) with even system size. In Ref. [4], while a two-fold ground state degeneracy is found for the usual 0° compactification along the vertical/horizontal lines of the square lattice in the symmetry sector $M_z = 0$ (one in $(+,+)$, one in $(-,-)$), compactifying the lattice at 45° indicates a non-trivial three-fold degeneracy in the symmetry sector $M_z = 0$ (one in $(+,+)$, $(+,-)$, $(-,+)$ each) for the uniform U1TC based on the QMC data. We investigate these findings of the uniform U1TC from the limit of isolated stars at $\lambda = 0$ in the following section.

## 3 Exactly solvable limit of checkerboard U1TC

Next, we show that the limiting case $\lambda = 0$ of the checkerboard U1TC is exactly solvable. It thus serves as a suitable starting point for degenerate perturbation theory. The approach taken here is therefore similar to the one used to study the fracton phase in the checkerboard XYTC [16]. This allows us to understand the compactification dependence of the ground-state degeneracy. We demonstrate the absence of topological order in the checkerboard U1TC for small $\lambda$ and discuss the different types of excitations on stars and plaquettes in this $\lambda$-regime in terms of confined fracton excitations.

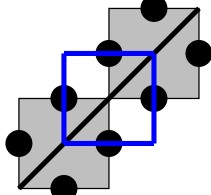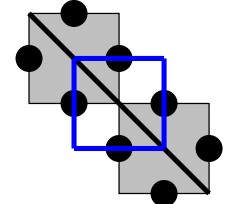

Figure 2: Constraint along the anti-diagonal (left) and diagonal (right) imposed by the condition $B_p |\Phi\rangle = |\Phi\rangle$ (acting on the blue plaquette $p$) on the local ground states of neighboring stars (shaded in gray). The black dots indicate the spin-1/2 degrees of freedom.

For $\lambda = 0$, the checkerboard U1TC Eq. (1) becomes a sum of local star operators that all commute with each other. Each operator $-\tilde{A}_{s_1}$ has three local ground states

$$|\boxslash\rangle = \frac{1}{\sqrt{2}}(|\bullet\!\!\:\substack{\circ\\ \bullet}\!\!\:\circ\rangle + |\circ\!\!\:\substack{\bullet\\ \circ}\!\!\:\bullet\rangle), \tag{5}$$

$$|\boxbackslash\rangle = \frac{1}{\sqrt{2}}(|\bullet\!\!\:\substack{\bullet\\ \circ}\!\!\:\circ\rangle + |\circ\!\!\:\substack{\circ\\ \bullet}\!\!\:\bullet\rangle), \tag{6}$$

$$|\boxempty\rangle = \frac{1}{\sqrt{2}}(|\bullet\!\!\:\substack{\circ\\ \circ}\!\!\:\bullet\rangle + |\circ\!\!\:\substack{\bullet\\ \bullet}\!\!\:\circ\rangle) \tag{7}$$

with eigenvalue $-1$, where the two colors of the dots visualizes the two states $|\!\uparrow\rangle$, $|\!\downarrow\rangle$ of each spin $1/2$. Further, there are ten excited states with eigenvalue $0$

$$|\bullet\!\!\:\substack{\circ\\ \circ}\!\!\:\circ\rangle, \quad |\circ\!\!\:\substack{\bullet\\ \circ}\!\!\:\circ\rangle, \quad |\circ\!\!\:\substack{\circ\\ \bullet}\!\!\:\circ\rangle, \quad |\circ\!\!\:\substack{\circ\\ \circ}\!\!\:\bullet\rangle, \quad |\bullet\!\!\:\substack{\bullet\\ \bullet}\!\!\:\circ\rangle, \quad |\bullet\!\!\:\substack{\circ\\ \bullet}\!\!\:\circ\rangle, \quad |\circ\!\!\:\substack{\bullet\\ \bullet}\!\!\:\bullet\rangle, \quad |\bullet\!\!\:\substack{\circ\\ \bullet}\!\!\:\bullet\rangle, \tag{8}$$

$$|\bullet\!\!\:\substack{\bullet\\ \bullet}\!\!\:\bullet\rangle, \quad |\circ\!\!\:\substack{\circ\\ \circ}\!\!\:\circ\rangle, \tag{9}$$

and three excited states with eigenvalue $+1$

$$\frac{1}{\sqrt{2}}(|\bullet\!\!\:\substack{\circ\\ \bullet}\!\!\:\circ\rangle - |\circ\!\!\:\substack{\bullet\\ \circ}\!\!\:\bullet\rangle), \quad \frac{1}{\sqrt{2}}(|\bullet\!\!\:\substack{\bullet\\ \circ}\!\!\:\circ\rangle - |\circ\!\!\:\substack{\circ\\ \bullet}\!\!\:\bullet\rangle), \quad \frac{1}{\sqrt{2}}(|\bullet\!\!\:\substack{\circ\\ \circ}\!\!\:\bullet\rangle - |\circ\!\!\:\substack{\bullet\\ \bullet}\!\!\:\circ\rangle). \tag{10}$$

We use the graphical notation introduced in Eqs. (5) to (7) to denote the three local ground states on stars $s_1$. In the absence of plaquette operators, one therefore has a ground-state degeneracy of $3^{N_{s_1}}$, where $N_{s_1}$ is the number of stars $s_1$ on the checkerboard lattice. Each global ground state $|gs\rangle$ corresponds to a product state of local superpositions of the three local ground states on each star.

## 3.1 Ground-state manifold and compactification

Next, we discuss the ground-state degeneracy in the exactly solvable limit and its dependence on the compactification. The presence of the term $-\sum_p B_p$ in the checkerboard U1TC introduces geometric constraints on the local ground states and leads to a reduction of the ground-state degeneracy. Indeed, the requirement that all conserved eigenvalues of plaquette operators are $b_p = +1$ for all $p$ in the ground state is not in line with all combinations of local ground states on neighboring stars. This is illustrated in Fig. 2. For example, if the local ground state $|\boxslash\rangle$ ($|\boxbackslash\rangle$) is chosen on one tile, then the top-right neighboring tile is restricted to be in the same state $|\boxslash\rangle$ ($|\boxbackslash\rangle$). By induction, we find that the entire anti-diagonal is restricted to the same local ground state $|\boxslash\rangle$ ($|\boxbackslash\rangle$). The same holds for any (anti-)diagonals. Therefore, we conclude that the ground-state degeneracy is reduced to a subextensive level, which we discuss in greater detail in the following.

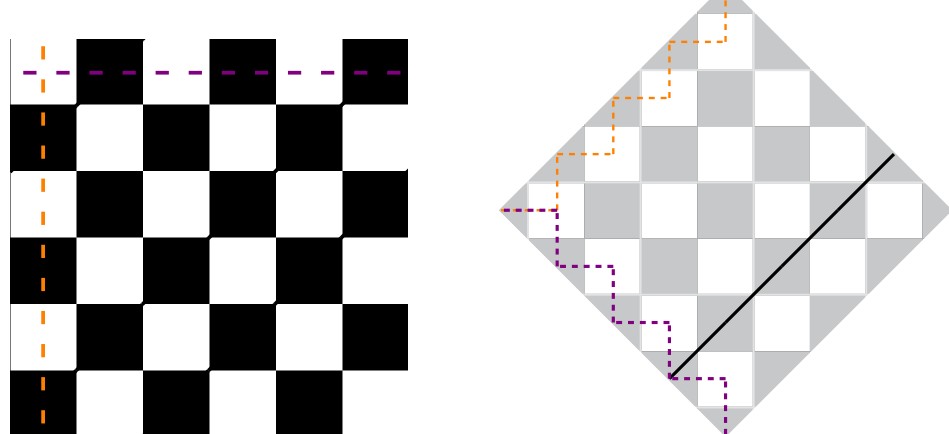

Figure 3: Left: Example of a global ground state for $\lambda = 0$ in the $0°$ compactification with $L = 6$. The state lies in the $(+, +)$ symmetry sector. Right: Example of a global ground state for $\lambda = 0$ in the $45°$ compactification with $L = 4$. The state lies in the $(-, +)$ symmetry sector.
Two non-contractible loops are shown by dashed lines.

We begin with the $0°$ compactification. The ground-state degeneracy is given by $2^{L/2+1}-1$, because if one anti-diagonal is constrained to be in the local ground state $|\diagup\rangle$, then both, the corresponding anti-diagonal has to be in the same local ground state as well as no other star can be realized in the diagonal local ground state $|\diagdown\rangle$ and vice versa. For the $45°$ compactification, we find $2^{L+1}-1$ ground states, following the same line of argument. The different ground-state degeneracies are thus explained by the different numbers of diagonals and anti-diagonals depending on the compactification. The ground states in the $0°$ compactification all lie in the symmetry sector $M_z = 0$ and are characterized by $(w_1, w_2) = (+1, +1)$ or $(-1, -1)$ for even system size because every diagonal and anti-diagonal crosses both $\mathcal{C}_1, \mathcal{C}_2$ once (see Fig. 3). In Ref. [4], the ground states of the uniform U1TC for $\lambda = 1$ were found in the same symmetry sectors. For the $45°$ compactification, the ground states also have $M_z = 0$ but lie in the sectors $(+1, +1)$, $(+1, -1)$, or $(-1, +1)$ for even system sizes, as the (anti-)diagonals cross only $(\mathcal{C}_1)$ $\mathcal{C}_2$. This matches again the symmetry sectors in which the ground states of the uniform U1TC were found according to the QMC calculations. Fig. 3 shows examples for ground states in both compactifications.

Starting from the exactly solvable limit $\lambda = 0$, we apply degenerate perturbation theory to find the ground state for $\lambda \ll 1$. We observe that the sub-extensive number of ground states only couple at order $L$ in perturbation theory on a finite cluster of extension $L \times L$ for all compactifications, so that the effective low-energy Hamiltonian remains exactly diagonal in the chosen basis up to arbitrary order in the thermodynamic limit. The same was encountered in the checkerboard XYTC [16].

However, while the subextensive ground-state degeneracy is conserved in the XYTC due to subsystem symmetries, the degeneracy is fully broken in fourth-order perturbation theory for the checkerboard U1TC (see App. A for details). The unique ground state is found to be the one where all local ground states are in the configuration $|\square\rangle$. The gap between this ground state with energy $E_{\text{gs}}$ and a state with one diagonal restricted to $|\diagdown\rangle$ with energy $E_1$ is given by

$$E_1 - E_{\text{gs}} = L\left[\frac{15}{28672}\lambda^4 + \mathcal{O}\left(\lambda^6\right)\right] \tag{11}$$

and grows subextensively with the linear system size $L$. Due to symmetry, the state with one anti-diagonal yields the same gap. In this order the difference in energy scales linearly with

the number of (anti-)diagonals.

For all compactifications, the unique ground state has therefore no long-range entanglement and is not topologically ordered. If this phase extends to $\lambda = 1$, this would contradict the previously suggested topologically ordered ground state for the uniform U1TC [4].

## 3.2 Excitations

In this subsection we discuss excitations on stars and plaquettes in the exactly solvable limit $\lambda \ll 1$. As we do not find a topologically ordered ground state in this regime, the low-energy excitations must be topologically trivial in the thermodynamic limit. Interestingly, we find that the excitations of the checkerboard U1TC correspond to confined fracton excitations, i.e., some excitations are fractons with restricted mobility, which is similar to the excitations in the XYTC [16] where fractonic excitations are protected by subsystem symmetries. However, the excitation energy of fractons or topologically non-trivial composites of them are shown to scale sub-extensively with the linear system size in the checkerboard U1TC in agreement with the absence of topological order in the thermodynamic limit.

### 3.2.1 Star excitations

First, let us focus on local excitations of stars $s_1$. We will only discuss local excitations with eigenvalue 0, as those with eigenvalue $+1$ do not possess any interesting properties and play no role at low energies. We find that such excitations with eigenvalue 0 exhibit restricted mobility in the subspace of $b_p = +1$ for all $p$, i.e., individual excitations are immobile type-1 fractons while composites can display dimensional reduction in terms of lineons. Fractons and lineons are found to have an excitation energy which scales sub-extensively with system size so that fractonic excitations are confined in the thermodynamic limit.

To see the restricted mobility of fractons and fracton composites, we analyze such excitations on top of the unique ground state in the regime $\lambda \ll 1$ (see also Fig. 4). Assuming that eigenvalues of all plaquette operators are $b_p = +1$, a local star excitation with eigenvalue 0 enforces that two (Eq. (8)) or four (Eq. (9)) of the surrounding stars must be in the local groundstate $|\boxbslash\rangle$, $|\boxslash\rangle$ or in an excited state with local eigenvalue 0. In analogy to the argument used for the ground-state manifold, this constraint extends along the entire (anti-)diagonal, forcing it to be in a specific ground-state configuration until the diagonal meets an excitation. An isolated star excitation is therefore attached to constrained (anti-)diagonals and cannot move without creating excitations by acting with operators with finite extent. These immobile excitations are called fractons.

In the following, we first discuss star excitations as defined in Eq. (8) in more detail. Single fractons can only be created for open boundary conditions, as shown in Fig. 4(a). Two-fracton composites can solely be formed for open boundary conditions or for the $0°$ compactification. For open boundary conditions, such composites connected by a line of $|\boxbslash\rangle$ or $|\boxslash\rangle$ states can move but only in one dimension orthogonal to the line, which is typically called a lineon excitation (see Fig. 4(b)). For the $0°$ compactification, two-fracton composites can only be created as illustrated in Fig. 5. Note that this composite is not a lineon because both fractons are immobile in the thermodynamic limit. Finally, composites of four fractons in rectangular configurations are mobile in two dimensions by acting with a finite-extent operator and are topologically trivial (see Fig. 4(c)). They exist for open boundary conditions and in every compactification.

Next, we discuss star excitations of the form $|\overset{\circ}{\underset{\circ}{\circ \, \circ}}\rangle$ or $|\overset{\bullet}{\underset{\bullet}{\bullet \, \bullet}}\rangle$ (see Eq. (9)). They can exist for open boundary conditions and for the $45°$ compactification as illustrated in Fig. 5. In contrast, for the $0°$ compactification, the constraint diagonals would intersect, prohibiting isolated excitations of this type.

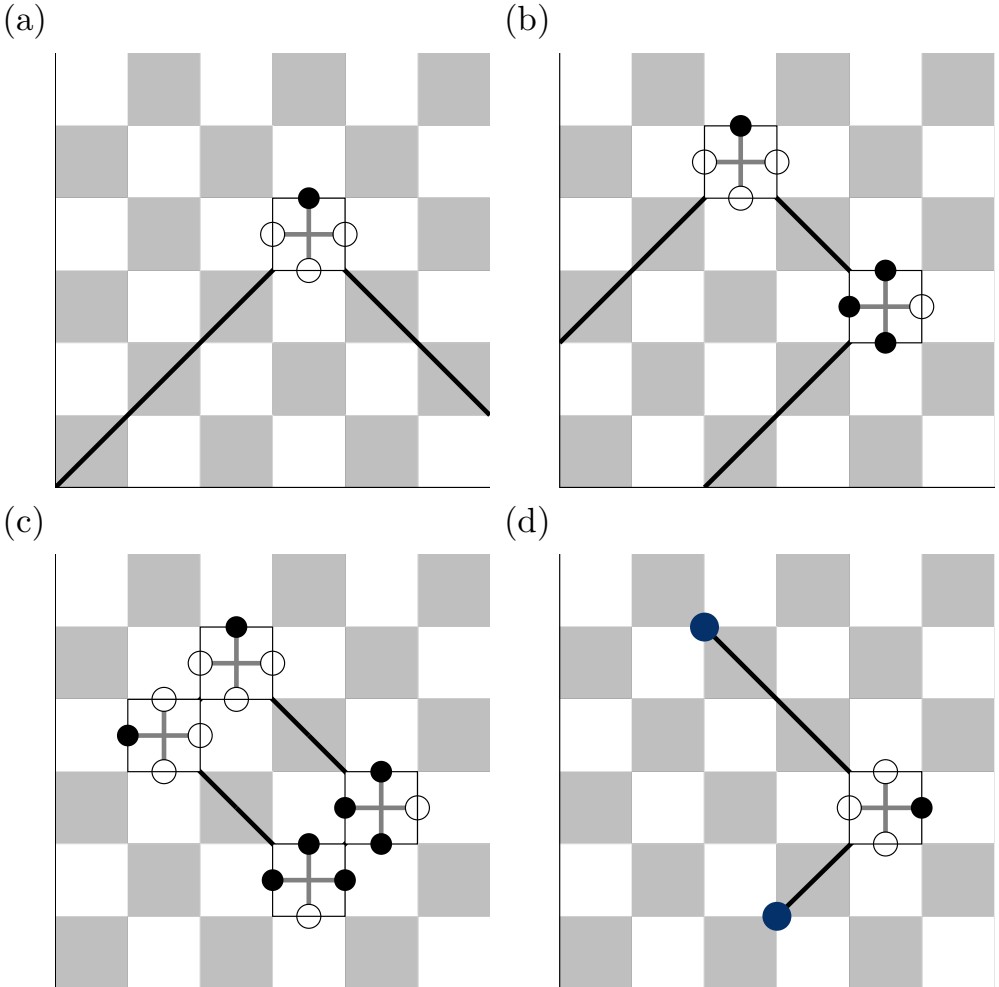

Figure 4: (a) Isolated fracton on a star with open boundary conditions. Due to the energy cost of constrained (anti-)diagonals, the excitation energy grows subextensively with system size $L$. (b) Excitation with two fractons on stars under open boundary conditions. The two fractons can be moved parallel to the two semi-infinite constrained diagonals. Due to the energy cost of these diagonals, the excitation energy grows subextensively with system size $L$. (c) Four-star excitations on the edges of a rhombus in the symmetry sector. The bound state of four excitations can move in two dimensions. (d) Two plaquette lineon excitations (large circles) and a fracton on a star. The lineons cannot move orthogonal to the diagonal of $|\diagdown\rangle$ or $|\diagup\rangle$ states without creating fracton excitations on stars.

Note that in all cases each local star excitation can also be replaced by their inverse, i.e., flipping all four spins.

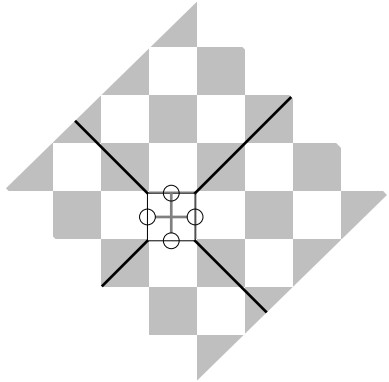 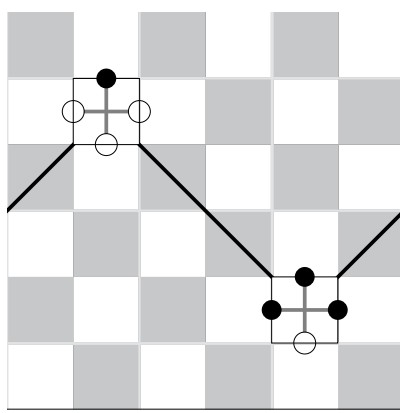

Figure 5: Left: In the 45° compactification, an isolated fracton as shown can be created on a star. Due to the infinite (anti-)diagonals, the energy of this state grows subextensively. Right: In the 0° compactification, two isolated fractons as shown can be created on stars. Again, due to the infinite (anti-)diagonals, the energy of this state grows subextensively.
Note that in all cases each local star excitation can also be replaced by their inverse, i.e., flipping all four spins.

Finally, let us clarify the fate of these excitations in the thermodynamic limit. The low-lying excitations are the rectangular four-fracton composites displayed in Fig. 4(c). In contrast, the energy gap of all other topologically non-trivial excitations grows subextensively with system size $L$. This is grounded in the finding in Subsec. 3.1 that added (anti-)diagonals increase the energy subextensively. All these excitations are therefore confined in the thermodynamic limit, but can influence the low-energy physics of finite systems significantly. As shown in Sec. 3.1 the ground state of the checkerboard U1TC does not display topological order for $\lambda \ll 1$ and the true low-energy excitations in the thermodynamic limit are topologically trivial.

### 3.2.2 Plaquette excitations

In addition to the star excitations, plaquette excitations corresponding to eigenvalues +1 appear at the open ends of (anti-)diagonal lines. These excitations can move only along the (anti-)diagonal directions without creating additional fracton excitations on stars as shown in Fig. 4(d) and are therefore classified as lineons. Similar to the star excitations, these lineons are also confined due to the presence of a connecting string of excited states. Moving a lineon away from its partner increases the energy linearly with distance, making isolated motion energetically costly. Note that two plaquette excitations placed on a (anti-)diagonal line represent the topologically trivial excitation in the thermodynamic limit.

## 4 Towards the uniform U1TC

In the last section we have shown that the checkerboard U1TC for $\lambda \ll 1$ displays no topological order and features a compactification-dependent ground-state degeneracy due to geometric constraints originating from the plaquette operators. Here we use exact diagonalization (ED) on a $4 \times 4$ system as well as high-order series expansions to explore the checkerboard U1TC on the full parameter line $\lambda \in [0, 1]$ for the 0° compactification. Most importantly, we

are interested to connect our results at small $\lambda$ to the numerical findings by finite-size and finite-temperature QMC simulations for $\lambda = 1$ [4].

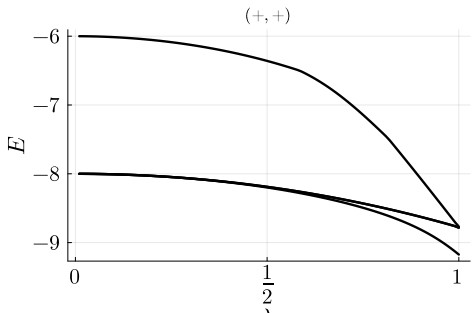 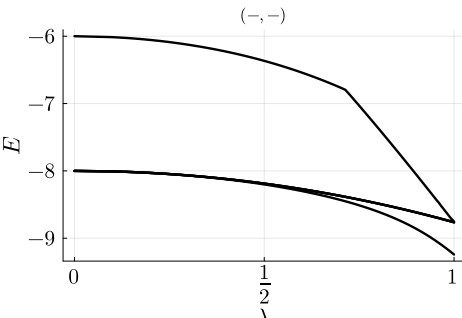

Figure 6: Low-energy levels from ED of a $4\times4$ cluster in the $0°$ compactification. Left: Four lowest energy levels in the $(+,+)$ symmetry sector. The first excited state is two-fold degenerate. Right: Four lowest energy levels in the $(-,-)$ symmetry sector. The first excited state is three-fold degenerate. The subextensive gap does not open at this system size. No higher-energy states for small $\lambda = 0$ approach the ground state on the full parameter line $\lambda \in [0,1]$.

Using ED on the $4 \times 4$ cluster, we compute the lowest energy eigenvalues in the symmetry sectors of the ground state at $\lambda = 0$ in the $0°$ compactification, namely $(+,+)$ and $(-,-)$, for varying $\lambda$ plotted in Fig. 6. Due to the finite size of this cluster, different $\lambda = 0$ ground states already couple at fourth order in perturbation theory in $\lambda$. As a consequence, the subextensive gap characteristic for large $L$ cannot be observed for small $\lambda$. Most importantly, there is no indication of a quantum phase transition on the full parameter line $\lambda \in [0,1]$.

Next, we estimate the energy gap between the global ground state with energy $E_{gs}$ and a state with one diagonal constrained in the state $|\boxslash\rangle$ with energy $E_1$ for the uniform U1TC in order to compare with the numerical findings of QMC simulations on finite clusters up to $L = 10$ [4]. The energy gap can be determined by high-order series expansions up to order 8, which is valid for finite systems with $L > 9$. The state with one diagonal constrained is part of the ground-state manifold in the $(-,-)$ symmetry sector for $\lambda = 0$. Different ground states do not couple up to order 8 for $L > 9$ so that $E_1$ and $E_{gs}$ can be calculated by corresponding expectation values. The monotonic series for this gap depending on $L$ is given by

$$E_1 - E_{gs} = L\left(0.00052315848625000013\lambda^4 + 0.000298375\lambda^6 + 0.000175125\lambda^8 + \mathcal{O}(\lambda^{10})\right). \tag{12}$$

For system size $L = 10$ and $\lambda = 1$ the gap is 0.00997 (0.00822) using the bare order-8 (order-6) series. Interestingly, the value for this energy gap is quite close to the one calculated with QMC for the same system size [4]. It is therefore likely that this gap, although infinite in the thermodynamic limit, was actually not distinguishable within the finite-temperature QMC simulations from a finite-size effect, that lead to the interpretation of unconventional ground-state degeneracy.

## 5 Conclusion

We have investigated the checkerboard U1TC. As for the XYTC [16], the limit of isolated star sublattices is exactly solvable and allows precise findings. We demonstrate that one can naturally explain the dependence of the ground-state degeneracy on the compactification by geometrical constraints of the plaquette operators. Further, fourth-order degenerate perturbation theory results in a non-degenerate unique ground state that does not display topological order.

High-order series expansions and exact diagonalization on small clusters do not give any evidence for a quantum phase transition up to the uniform U1TC investigated in Ref. [4]. At the same time, energy gaps are shown to be extremely small up to the uniform U1TC. These small energy scales can be traced back to a large confinement scale of fracton excitations which cannot exist as single low-energy excitation in the thermodynamic limit but only as topologically trivially composite particles. Such gaps are not unambiguously detectable by recent finite-size and finite-temperature QMC simulations in the uniform limit [4]. Consequently, our findings point towards the absence of topological order in the checkerboard U1TC along the full parameter line including the uniform case.

Furthermore, it would be interesting to investigate the relation between the confined fracton picture introduced in this work and the Hilbert space fragmentation found in Ref. [4] and the Ising model in a weak transverse field [6,7].

**Note added**.– While finalizing this work, we became aware of a related independent study [17], which is expected to appear on arXiv concurrently.

## Acknowledgements

We thank Simon Trebst, Jiaxin Qiao, and Yoshito Watanabe for fruitful discussions.

**Author contributions**     MV: Conceptualization, Data curation, Formal analysis, Investigation, Methodology, Visualization, Writing – original draft, Writing – review & editing. VK: Conceptualization, Supervision, Writing – review & editing. LL: Conceptualization, Supervision, Writing – review & editing. AS: Conceptualization, Supervision, Writing – review & editing. KPS: Conceptualization, Funding acquisition, Methodology, Resources, Supervision, Writing – original draft, Writing – review & editing.[1]

**Funding information**     KPS gratefully acknowledge financial support by the Deutsche Forschungsgemeinschaft (DFG, German Research Foundation) through the TRR 306 QuCoLiMa ("Quantum Cooperativity of Light and Matter") - Project-ID 429529648 (KPS). KPS acknowledges further financial support by the German Science Foundation (DFG) through the Munich Quantum Valley, which is supported by the Bavarian state government with funds from the Hightech Agenda Bayern Plus.

## A    Fourth-order perturbation theory about the isolated star limit

In this appendix we apply the degenerate-perturbation formalism of Takahashi [18] to the checkerboard U1TC. Let $P$ be the projector onto the unperturbed ground-state manifold for $\lambda = 0$ and

$$S = \frac{1-P}{E_0 - H}, \tag{A.1}$$

where $E_0$ is the unperturbed ground-state energy and $H$ the Hamiltonian from Eq. (1). Using Takahashi's method the effective Hamiltonian up to order $\lambda^4$ reads

$$H_{\text{eff}} = H_0 + \lambda H_1 + \lambda^2 H_2 + \lambda^3 H_3 + \lambda^4 H_4, \tag{A.2}$$

---

[1]Following the taxonomy CRediT to categorize the contributions of the authors.

with

$$H_1 = PVP \,, \tag{A.3}$$

$$H_2 = PVSVP \,, \tag{A.4}$$

$$H_3 = PVSVSVP + PVPVSSVP + PVSSVVP \,, \tag{A.5}$$

$$
\begin{aligned}
H_4 = {} & \frac{1}{2} PVPVPVSSSVP - \frac{1}{2} PVPVSVSSVP - \frac{1}{2} PVPVSSVSVP - \frac{1}{2} PVSVPVSSVP \\
& + PVSVSVSVP + \frac{1}{2} PVSSSVPVPVP - \frac{1}{2} PVSSVSVPVP - \frac{1}{2} PVSSVPVSVP \,.
\end{aligned}
\tag{A.6}
$$

Operator expressions of higher orders can be easily generated model-independently.

## A.1   Action of $\tilde{A}_s$ on the ground-state manifold

Acting with a single star operator $\tilde{A}_s$ on a product state either flips the four spins on that star or annihilates the state. For the four neighboring stars let

$$|x_1, x_2, x_3, x_4\rangle = |\tfrac{a_1+b_1}{\sqrt{2}}, \tfrac{a_2+b_2}{\sqrt{2}}, \tfrac{a_3+b_3}{\sqrt{2}}, \tfrac{a_4+b_4}{\sqrt{2}}\rangle \,, \tag{A.7}$$

where $x_i$ is a generic ground state, $a_i \in \{|{\bullet}{\circ}{\circ}\rangle, |{\bullet}{\bullet}{\circ}\rangle, |{\bullet}{\circ}{\bullet}\rangle\}$ and $b_i = \tilde{A}_i a_i$. Because $\tilde{A}_s$ flips or annihilates product states, its action always produces four excitations on the surrounding local ground states. Hence, on an infinite lattice only even orders in perturbation contribute: every excited star must be acted on a second time before $P$ can project back into the ground-state subspace. Consequently, all terms containing a factor $PVP$ vanish in the thermodynamic limit.

## A.2   Second-order contribution

For the generic configuration $|x_1, x_2, x_3, x_4\rangle$ a straightforward calculation gives

$$\langle x_1, x_2, x_3, x_4 | \tilde{A}_s S \tilde{A}_s | x_1, x_2, x_3, x_4 \rangle = -\frac{6}{2^6} \,. \tag{A.8}$$

The result is independent of which local ground states are present on the neighboring stars.

## A.3   Fourth-order processes

All fourth-order diagrams that contribute differently to ground states with or without constrained (anti-)diagonals are grouped into two classes, illustrated in Fig. 7.

Only the operator sequence $PVSVSVSVP$ contributes differently to ground states with and without a ▨-restricted diagonal.

For the state $|\square, \square, \boxslash, \boxslash, \square, \square\rangle$ shown on the left of Fig. 7.

$$\langle \square, \square, \boxslash, \boxslash, \square, \square | P\tilde{A}_s S \tilde{A}_s S \tilde{A}_s S \tilde{A}_s P | \square, \square, \boxslash, \boxslash, \square, \square \rangle = -\frac{1}{6} \frac{1}{4^2} \frac{10}{2^6} \,. \tag{A.9}$$

For $|\square, \square, \square, \square, \square, \square\rangle$ the same process gives

$$
\begin{aligned}
& \langle \square, \square, \square, \square, \square, \square | P\tilde{A}_s S \tilde{A}_s S \tilde{A}_s S \tilde{A}_s P | \square, \square, \square, \square, \square, \square \rangle \\
& = -\frac{1}{4^2} \frac{1}{2^6} \left[ \frac{4}{4} + \frac{4}{8} + \frac{1}{4}\left(\frac{2}{4} + \frac{2}{8} + \frac{4}{6}\right) \right].
\end{aligned}
\tag{A.10}
$$

For processes shown on the right of Fig. 7 we find, e.g.,

$$\langle \square, \square, \square, \square, \square, \square, \square | P\tilde{A}_s S \tilde{A}_s S \tilde{A}_s S \tilde{A}_s P | \square, \square, \square, \square, \square, \square, \square \rangle = -\frac{18}{4^2} \frac{1}{7} \frac{1}{2^7} \,, \tag{A.11}$$

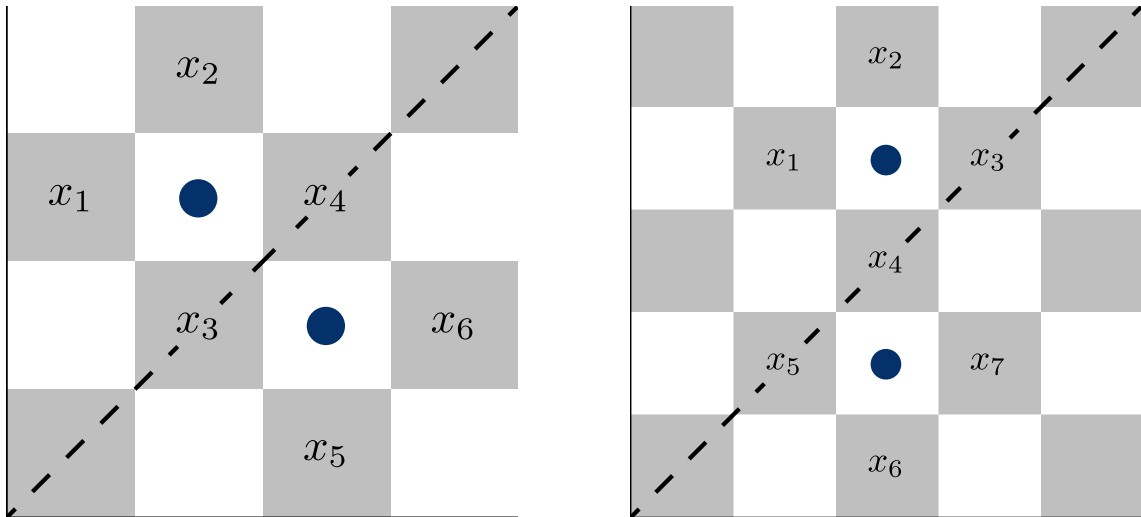

Figure 7: Fourth-order process crossing a constraint diagonal. The $x_i$ indicate the ordering convention used in the calculation. The blue dots show the stars on which the $\tilde{A}_s$ perturbation acts twice in fourth order. The dashed line indicates the position of the diagonal for the $E_1$ calculation.

while a configuration with a $|\boxslash\rangle$ diagonal yields

$$\langle \boxempty, \boxempty, \boxslash, \boxslash, \boxslash, \boxempty, \boxempty | P\tilde{A}_s S\tilde{A}_s S\tilde{A}_s S\tilde{A}_s P | \boxempty, \boxempty, \boxslash, \boxslash, \boxslash, \boxempty, \boxempty \rangle = -\frac{1}{4^2}\frac{1}{2^7}\frac{1}{2}\left(\frac{18}{6} + \frac{18}{8}\right). \tag{A.12}$$

In the following we compare the energy $E_1$ with one diagonal $|\boxslash\rangle$ to the energy $E_{\mathrm{gs}}$ with no (anti-)diagonals. Summing all non-cancelling fourth-order contributions gives

$$\Delta E = E_1 - E_{\mathrm{gs}} = 4L\Big[\frac{1}{4^2 2^6}\Big(\frac{4}{4} + \frac{4}{8} + \frac{1}{4}(\frac{2}{4} + \frac{2}{8} + \frac{4}{6})\Big) - \frac{1}{6}\frac{10}{4^2 2^6}$$
$$+ \frac{18}{4^2}\frac{1}{7}\frac{1}{2^6} - \frac{1}{4^2}\frac{1}{2^6}\frac{1}{2}\Big(\frac{18}{6} + \frac{18}{8}\Big)\Big]\lambda^4 + \mathcal{O}(\lambda^6) \tag{A.13}$$
$$\approx 5.232 \times 10^{-4}\, L\, \lambda^4.$$

Because $\Delta E > 0$, the fully $|\boxempty\rangle$ state is the unique ground state up to this order; the gap scales linearly with $L$ and hence diverges in the thermodynamic limit, albeit sub-extensively.

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
