# Peer review of "Absence of topological order in the $U(1)$ checkerboard toric code"

_SciPost Physics_

## Round 2 · Referee Report · Helene Spring (Referee 1) · 2025-8-12

Report

This manuscript addresses a recent claim [Ref. 4] that the uniform U(1)-enriched toric code (U1TC) may host a non-Abelian topologically ordered phase. That claim was based primarily on quantum Monte Carlo simulations showing compactification-dependent ground-state degeneracies and apparent UV/IR mixing. The present work presents an analytical and numerical investigation of the same model, with the goal of assessing whether the U1TC exhibits topological order. Their analysis reveals that the behavior observed in prior numerical work can instead be explained as finite-size and geometric effects in a trivial phase. This makes the current work both timely and highly relevant to the field of topologically ordered matter.

One of the most important results concerns the compactification dependence of the ground-state degeneracy. The authors show that, in the λ → 0 limit, the degeneracy scales sub-extensively and depends explicitly on the geometry of the torus (e.g., 0° vs 45° compactification). They track how this dependence arises from geometric constraints enforced by the plaquette terms, rather than from topological sector counting.
This result directly challenges the interpretation of compactification-dependent degeneracy in Ref. 4 as a signature of UV/IR mixing or non-Abelian topological order. Instead, the degeneracy is shown to be lifted at fourth order in perturbation theory, revealing a topologically trivial ground state in the thermodynamic limit.

The low-energy excitations of the checkerboard U1TC are also analyzed in detail. While some excitations resemble fractons, they are found to be confined. Only topologically trivial bound states of multiple excitations remain viable in the thermodynamic limit. This further confirms the absence of deconfined anyonic excitations and supports the conclusion that the system is in a trivial phase.

The manuscript plays a critical clarifying role in the ongoing discourse about the nature of symmetry-enriched toric codes and potential realizations of non-Abelian topological order in two-dimensional models. By rigorously analyzing the U1TC using perturbative and numerical tools, the authors effectively refute the claim of topological order in this system and demonstrate that the phenomena seen in Ref. 4 are likely finite-size effects rather than signatures of a non-trivial phase.
The work is clearly written, well-structured, and based on solid methodology. It will be of interest not only to researchers working on toric codes and fractons, but also to those studying Hilbert space fragmentation, quantum simulation platforms, and the broader class of symmetry-enriched topological phases.

⠀I recommend acceptance, possibly with minor revisions, focused primarily on addressing the points below. The paper is a substantial and well-executed contribution to the literature and provides an important corrective to recent claims in the field.

Requested changes

I have some suggestions for improvement that I would like the authors to consider. * While the paper convincingly shows that the system is gapped and trivial, the magnitude of the gap remains small, even at λ = 1. It would be valuable to include a brief discussion of whether (and how) the gap could be made larger by adding symmetry-allowed perturbations. This might help distinguish the trivial phase more sharply from the finite-size effects that complicate numerical studies. This would also show that numerical studies alone could detect the triviality due to the presence of a gap. * The authors mention that their series expansion results agree with QMC gaps at λ = 1. Including a figure or table comparing these values explicitly could strengthen this point.

Recommendation

Publish (meets expectations and criteria for this Journal)

---

## Round 2 · Referee Report · Anonymous (Referee 2) · 2025-9-1

Report

In this work, the authors analyze a two-dimensional spin model recently introduced by Wu et al in Ref 4. This "U1TC" model is U(1)-symmetrized version of the square lattice toric code. In Ref 4 it was claimed that this model has topological ground state degeneracy which moreover depends on precise boundary conditions. One potential resolution which was offered was that the phase of matter is described by non-abelian topological order. In the present manuscript, Vieweg et al introduce a checkerboard version of the U1TC, which allows one to tune between the original model by Wu et al (lambda=1) and a solvable limit (lambda=0). The authors show that the solvable limit is in a trivial phase of matter (albeit with interesting short-distance non-universal properties) and they argue using small-scale ED and extensive series-expansion calculations that this trivial phase of matter persists to the lambda=1 limit, thereby arguing against some of the claims made in Ref 4.

I commend the authors on undertaking a critical analysis of claims made in the literature. I find their solvable lambda=0 limit quite insightful, and it nicely reproduces the observed near-ground-state-degeneracy in Ref 4 and explains that this does not require a non-trivial origin (i.e., it is consistent with a trivial phase of matter, since the authors explain that there is a nonzero gap between these ground states at fourth-order in lambda). While this is a nice consistent explanation, which is considerably simpler than the explanation offered in Ref 4, I found Section 4 (devoted to the lambda -> 1 case) a bit lacking to be entirely conclusive. In addition, while this is a nice careful analysis of a previously-introduced model, and while it is important to rectify unjustified claims in the literature, it is not clear to me that the particular insights gained in this work meet the 'groundbreaking' or 'opening new pathways' criteria listed for SciPost Physics. The manuscript probably does meet the criteria for SciPost Physics Core (although see the comments below). I do not directly see why it meets the criteria for SciPost Physics, but I remain open-minded for a potential rebuttal, based on the below comments. Overall, the content of the manuscript feels a bit minimal, and it would be nice if it went beyond merely rectifying previous work. The current take-away message I have as a reader is "the (checkerboard) U1TC is not so interesting, move on", which is of course solid work but does not evoke the feeling of a groundbreaking piece of work.

  • The ED discussion is quite brief (roughly a paragraph long) and is restricted to a 4x4 geometry (presumably 442 qubits, which is then reduced using quantum numbers such as U(1) conservation etc). I found certain claims a bit confusing such as "No higher-energy states for small λ = 0 approach the ground state on the full parameter line". If we are only given one fixed system size, it is nearly impossible to tell whether there is a trend of states approaching the ground state. Moreover, this section would benefit from a more detailed comparison to the numerical claims made in Ref 4. For instance, that work commented on the phase being gapped due to having a large gap. Can the authors of the present work compare their gap to the one reported in Ref 4?

  • On a related note: is there a reason the authors do not use a duality transformation to reduce the number of qubits? By applying the usual Kramers-Wannier transformation, one would obtain a model of qubits on the vertices of the square lattice, where the Hamiltonian would contain terms of the form "X", "ZXZ" and "ZZXZZ" (like the square lattice cluster model), and the U(1) symmetry is now generated by \sum Z_i Z_j. In this representation, the "4x4" lattice would only have 16 qubits, which is easily simulable even without quantum numbers. It thus seems likely one can push to larger system sizes. This might allow the authors to consider more than one system size, such that one can get a sense of the behavior of the gap in the thermodynamic limit et cetera.

  • The authors perhaps over-emphasize the non-abelian claims of Ref 4. If I take a cursory glance at Ref 4, it does not even mention "non-abelian" in the abstract, and the word only appears a total of three times in their manuscript. The first mention is on p6 of their published manuscript: "These arguments suggest a logical possibility that the U(1) toric code may realize non-Abelian topological order". This feels like a pretty mild claim, and I do not think that the non-abelian nature is really an explicit or main conjecture, it is merely offered as a logical explanation. More importantly, Ref 4 does clearly claim topological ground state degeneracy, so that feels like a safer point to argue against.

Recommendation

Accept in alternative Journal (see Report)

---

## Editorial Decision

resubmitted